# Fall Applications of Ethephon Modulates Gene Networks Controlling Bud Development during Dormancy in Peach (*Prunus Persica*)

**DOI:** 10.3390/ijms23126801

**Published:** 2022-06-18

**Authors:** Jianyang Liu, Md Tabibul Islam, Suzanne Laliberte, David C. Haak, Sherif M. Sherif

**Affiliations:** 1Alson H. Smith Jr. Agricultural Research and Extension Center, School of Plant and Environmental Sciences, Virginia Tech, Winchester, VA 22602, USA; liujy4@vt.edu (J.L.); tabibul@vt.edu (M.T.I.); 2School of Plant and Environmental Sciences, Virginia Tech, Blacksburg, VA 24061, USA; suzanne9@vt.edu (S.L.); dhaak@vt.edu (D.C.H.)

**Keywords:** peach, bud dormancy, ethylene, transcriptome, bloom

## Abstract

Ethephon (ET) is an ethylene-releasing plant growth regulator (PGR) that can delay the bloom time in Prunus, thus reducing the risk of spring frost, which is exacerbated by global climate change. However, the adoption of ET is hindered by its detrimental effects on tree health. Little knowledge is available regarding the mechanism of how ET shifts dormancy and flowering phenology in peach. This study aimed to further characterize the dormancy regulation network at the transcriptional level by profiling the gene expression of dormant peach buds from ET-treated and untreated trees using RNA-Seq data. The results revealed that ET triggered stress responses during endodormancy, delaying biological processes related to cell division and intercellular transportation, which are essential for the floral organ development. During ecodormancy, ET mainly impeded pathways related to antioxidants and cell wall formation, both of which are closely associated with dormancy release and budburst. In contrast, the expression of dormancy-associated MADS (DAM) genes remained relatively unaffected by ET, suggesting their conserved nature. The findings of this study signify the importance of floral organogenesis during dormancy and shed light on several key processes that are subject to the influence of ET, therefore opening up new avenues for the development of effective strategies to mitigate frost risks.

## 1. Introduction

Peach is an important tree fruit in temperate regions worldwide. In recent years, climate change is challenging peach production by increasing the occurrence of spring frosts. Spring frosts are freezing temperatures that occur after peach buds have begun to regrow and lost their cold hardiness. Devastating frost events can cause severe reduction in or complete loss of fruit yield, owing to the lack of effective frost mitigation strategies. In light of this, a frost-avoidance approach has been proposed to delay the bloom date until the risk of frost damage has passed or diminished. Previous studies have shown that fall-applied ethephon (2-chloroethylphosphonic acid), an ethylene-releasing plant growth regulator (PGR), can delay bloom in peach by a few days or even weeks [1], which can significantly reduce the risk of frost damage. However, ethephon (ET) application often causes mild to moderate injuries to peach trees, such as gummosis, floral bud death, and dieback of the branches, which limit the utility of ET. Our recent investigations have also indicated that ET may affect flowering phenology in peach through modulating the progression of bud dormancy, which is subject to the influences of phytohormone fluctuation, oxidative pressure, and carbohydrate metabolism [2,3]. An increased understanding of the mechanisms by which ET induces bloom delay is fundamental to developing effective strategies to combat spring frost and elucidate potential mechanisms of injury that can be ameliorated.

Bud dormancy in perennial species is a critical stage for surviving low winter temperatures. Winter dormancy can be divided into two sequential phases, endodormancy and ecodormancy, in which bud regrowth is inhibited by intrinsic signals and environmental constraints, respectively [4]. During endodormancy, buds remain unresponsive to favorable conditions and must experience a period of chilling temperature (chilling requirement or CR), before transitioning to ecodormancy. Temperatures in the range of 0–7.2 °C are most effective in satisfying CR and are widely used to quantify chilling accumulation [5,6]. In ecodormancy, bud growth is inhibited by unfavorable conditions, such as low temperatures or a short photoperiod, and some time in warmer temperatures (heat requirement or HR) is required for buds to proceed to budburst and flowering. HR is commonly expressed as growing degree hours (GDH), registering growth-conducive temperatures in the range of 4.5–22.5 °C [7,8]. Both CR and HR in Prunus are genetically controlled [9,10] and are a critical determinant of climatic distribution and adaptation of peach genotypes. Despite the distinct requirements for releasing the endodormancy and ecodormancy, no reliable phenotypic or molecular markers have been identified to distinguish the two stages.

Dormancy is a highly complex process that involves interactions between several internal signals and environmental cues. Unlike many other deciduous species which have a true “resting” period during dormancy, dormancy in Prunus is characterized by ongoing metabolic activities, through which the floral buds undergo continuous and progressive morphogenesis before flowering [11,12]. Such developmental events rely on the orchestration of multiple internal processes, including carbohydrate metabolism, hormone action, reactive oxidative species (ROS) homeostasis, cell division, differentiation, and enlargement. Carbohydrate metabolism during dormancy provides both osmotic control in cold acclimation and the trophic control of bud growth. In peach buds, an increase in soluble carbohydrates was found to correlate positively to the rate of budbreak [13]. Recently, Fadon et al. [14] demonstrated that the starch levels in the primordia of sweet cherry (*P. avium*) increase with the chilling accumulation, culminating in the fulfillment of CR. Similarly, Hernandez et al. [15] indicated peach buds accumulate starch in cold conditions, in addition to accumulating soluble sugars. These studies support the hypothesis that budbreak capacity is marked by the bud’s ability to acquire and utilize soluble carbohydrates [13].

As natural byproducts of normal oxygen metabolism, reactive oxygen species (ROS) tend to accumulate during dormancy due to cold stress and reduced metabolic activity. Recent studies indicate the accumulation of ROS plays an important signaling role that drives the progression of dormancy and the eventual budbreak [3,16,17]. In Japanese pear (*P. pyrifolia*), hydrogen peroxide (H_2_O_2_) levels increase as endodormancy progresses, culminating in the endodormancy release [18], and an exogenous application of H_2_O_2_ can substitute for the chilling effect and promote budbreak (Kuroda et al., 2005). Further, the effect of hydrogen cyanamide (HC), a compound commonly used in stimulating dormancy breaking, has been attributed to its ability to induce a rapid accumulation of H_2_O_2_ [19]. Our previous study indicated that ET alters the dynamics of H_2_O_2_ and the superoxide anion radical (O_2_^•−^) in peach buds, leading to extended endodormancy and delayed bloom [3].

Bud dormancy is also under the tight regulation of plant hormones. Among them, abscisic acid (ABA) and gibberellin (GA) play the central roles, antagonistically regulating dormancy onset and release. High levels of ABA are associated with dormancy initiation and maintenance, in which ABA inhibits growth through processes such as the arrest of the cell cycle and the blockage of symplastic transport [20,21]. In contrast, the increase in GA toward the release of dormancy restores the intercellular communication, enabling the mobility of important nutrients and growth-promoting signals [21,22], and promotes growth through the deactivation of the DELLA proteins [23]. GA can also increase ROS levels and energy metabolism, both of which are critical for dormancy release [24,25]. As a stress-responsive hormone, ethylene is involved in both dormancy initiation and release, in which it interacts closely with ABA, GA and ROS [26,27]. Other hormones such as indole-3-acetic acid (IAA), cytokinin (CK), and jasmonic acid (JA) are also involved in dormancy, and their exact functions remain to be fully characterized.

At the early stage of dormancy establishment, the dormancy-associated MADS-BOX (DAM) genes are of particular importance. DAM genes were first identified in peach and now have been recognized as a master regulator of dormancy in many deciduous species. DAMs are six tandemly arranged genes, closely related to the floral-repressing gene SHORT VETETATIVE PHASE (SVP) in Arabidopsis [28]. Transgenic studies indicated that the overexpression of *PmDAM6* from Japanese apricot in poplar (*Populus trichocarpa*) and apple (*M. domestica*) accelerates growth cessation and terminal bud set, and it delays budbreak [29,30]. In peach, these genes were found to have distinct expression patterns, with *DAM1/2/4* coincident with growth cessation and bud formation, and *DAM5/6* accompanying dormancy transition [31]. In sweet cherry, *DAM1/5* are also involved in promoting flower development during dormancy, possibly through upregulating floral organ identity genes [32]. Molecular data indicate that DAM genes are controlled by transcriptional regulation and epigenetic modification, which is driven by chilling accumulation [33]. With the accumulating evidence, DAM genes appear to be cognate regulators of dormancy through integrating chilling perception and the downstream regulatory pathways.

Over the last decade, much progress has been achieved in elucidating the regulatory mechanisms underpinning bud dormancy, especially the upstream regulatory steps. Mounting evidence has converged to support a DAM-centered regulation model, in which low temperatures activate a C-binding factor (CBF), which directly upregulates DAM, thereby inducing growth cessation and dormancy establishment [32,34,35,36]. Relatively less characterized and downstream of this regulatory hub are the biological processes that facilitate budbreak and flowering. In the present study, ET-mediated bloom delay was used as a model to elucidate the regulatory pathways regulating dormancy and bloom time regulation in peach. To this end, the floral bud from ET-treated and control peach trees were collected during endodormancy and ecodormancy and an RNA-Seq approach was employed to investigate the transcriptomic changes associated with the bloom delay in peach.

## 2. Results

In this study, we first characterized the phenotypic responses of dormant buds to the ET treatment (Figure 1A–D). ET increased the CR and HR of the dormant buds by 131 CH and 901 GDH, respectively (Figure 1A). Two months after the ET treatment, prolific gummosis was observed on the tree barks on the ET-treated trees, but not on the control trees (Figure 1B). At 1100 CH, the floral organs became readily distinguishable in the control, while they were only emerging in the ET treatment (Figure 1C). After budburst, when the control trees reached full bloom, the ET treatment only started to flower (Figure 1D). To reveal the underlying mechanisms of the ET effects on the molecular level, we monitored the transcriptomic changes in the buds of the ET-treated and untreated trees at regular intervals based on chill and heat accumulation. These samples evenly reflected the molecular characteristics of the buds throughout the stages of endodormancy and ecodormancy. Among the five samplings, the first three (200, 600, and 1000 CH) were during the endodormancy stage, and the last two (1000 and 3000 GDH) were in the ecodormancy stage.

### 2.1. Transcriptomic Profiles Clustered Primarily Based on Dormancy Stages

To explore the overall transcriptomic profiles of the 30 sequenced bud samples (five time points, two treatments, and three biological replicates), the sample distances were calculated based on FPKM expression values and visualized using a principal component analysis (PCA) (Figure 2A). The first two principal components (PC) explained 48.5 and 17.1% of the total variance, respectively, classifying the 30 samples into five groups, each corresponding to a sampling time. PC1 clearly separated the 3000 GDH group from the other four groups, which were better separated by PC2. On PC2, the 200 CH group was discriminated from the other three groups, which were marginally overlapping. The overall group separation pattern indicated that the transcriptomic profiles at the early stage of endodormancy (200 CH) and the late stage of ecodormancy (3000 GDH) were more distinct from other stages in between. Within each group, there was no clear distinction between the ET treatment and control at 200 CH, 600 CH, and 1000 GDH, and there was only slight to moderate differentiation at 1000 CH and 3000 GDH.

To detect differentially expressed genes (DEG) between gene sets to be compared, we performed differential expression analysis using DESeq2 on the threshold of FPKM > 2 and adjusted P-adj < 0.05. In comparison between endodormancy and ecodormancy, regardless of treatments, a total of 8668 genes were identified as differentially expressed genes (DEGs) (Figure 2B), in which 3878 were downregulated and 4790 were upregulated in ecodormancy. Among the downregulated DEGs, the three most significantly downregulated genes were the encoding lipid-transfer protein (P. 5G119200), antifungal surface protein (P.1G133700), and keratin-associated protein (P.4G184400), whereas the three most significantly upregulated DEGs were the encoding proline-rich protein (P.2G139000), hydroxyproline O-galactosyltransferase (P.2G131800), and gibberellin-regulated protein (P.1G038000).

The numbers of the DEGs between the ET treatment and the control varied greatly at different stages of dormancy. At 200 and 600 CH, there were 7 and 111 DEGs, respectively, reflecting a relatively small transcriptional difference at the early stage of the dormancy. This difference increased at 1000 CH and 3000 GDH as the number of identified DEGs rose to 610 and 936, respectively, with a greater number of genes downregulated. Notably, there were only 237 DEGs at 1000 GDH. The differences in the DEGs across all sampling points was presented in Figure 2D. The majority of the DEGs were unique to each sampling time, with a small number of DEGs overlapped between two adjacent time points and, rarely, a few DEGs shared among three time points.

A hierarchical clustering analysis on all the samples produced five well-separated groups corresponding to the sampling time (Figure 3). The expression patterns in groups of endodormancy (200, 600, and 1000 CH) were in sharp contrast to those in ecodormancy (1000 and 3000 GDH), in which the genes that were upregulated during endodormancy became downregulated during ecodormancy, and vice versa. The difference in expression patterns between the ET treatment and the control increased over time, and the most distinct patterns were found in the 3000 GDH group.

### 2.2. Annotation of DEGs between ET Treatment and Control during Endodormancy

To reveal the biological functions and pathways in which the identified DEGs are involved, we performed Gene Ontology (GO) and KEGG pathway enrichment analyses. Using the threshold of P-adj < 0.05, no significant GO or KEGG terms were found from the DEGs identified between endodormancy and ecodormancy regardless of treatment, nor from the DEGs identified between the ET treatment and control across all sampling times. Therefore, we focused the functional analyses on the DEGs between the ET treatment and control at each time point.

No significant GO terms were generated at the 200 CH time point, likely because of the small number of DEGs identified. At 600 CH (Figure 4A), the downregulated DEGs were enriched in eight terms in molecular function (MF) and six in biological pathway (BP) categories. These terms comprised three main groups: (a) mitosis, which includes microtubule processes, tubulin binding, skeletal protein binding, and the movement of cell or subcellular components, which is featured by four genes encoding kinesins; (b) the saccharide metabolic process, including the biosynthesis of β-glucan/glucan, polysaccharide, cellulose, polysaccharides; and (c) the hydrolysis of pyrophosphates and acid anhydrides, catalyzed by pyrophosphatase (PPase) and acid anhydrides hydrolase (AAH), respectively.

At 1000 CH, the upregulated DEGs were associated with five terms, one related to DNA replication helicase, one in transmembrane transport, and three related to hydrolysis of ATP or ATP-like bonds, including PPase, AAH, and ATPase. The downregulated DEGs were enriched in five GO terms, and in the term of cell wall modification, most of the DEGs were related to pectinesterase, which catalyzes the degradation of pectin, a main component of cell walls.

When the expression levels of the DEGs from each GO cluster were profiled over the five time points using the mean z-scores, distinct patterns were observed between the ET treatment and the control as shown in Figure 4B–E. For the downregulated DEGs at 600 CH, the control showed a clearly defined peak at 1000 CH, while the ET treatment only showed a moderate increase (Figure 4B). For the upregulated DEGs at 1000 CH, the expression of the ET treatment peaked at 1000 CH, whereas the control peaked at 600 CH, which was lower than the ET peak (Figure 4C). Such a peak-shifting pattern was also observed in the downregulated DEGs from 1000 CH, with the ET treatment peaking at 1000 GDH and the control peaking at 1000 CH (Figure 4D).

It is noteworthy that two GO terms (hydrolase activity, acting on acid anhydrides and pyrophosphatase activity) were generated in both the downregulated DEGs of 600 CH and upregulated DEGs of 1000 CH (Figure 4A). Their mean expression patterns were highly similar to that of the upregulated DEGs from 1000 CH (Figure 4E), indicating these genes were representative of this cluster.

### 2.3. Validation of the ABC Transporter Genes That Were Responsive to ET Treatment

DEGs enriched in the PPase and AAH clusters were differentially regulated by ET in two sequential time points, suggesting they are worth further examination as candidate ET-responsive genes in regulating endodormancy. These two clusters contained 21 enriched genes, in which 10 were related to the ABC transporters, belonging to four subfamilies: B, C, G, and pleiotropic drug resistance (PDR). To further analyze the expression of these genes, RT-qPCR was performed to validate the expression observed in the RNA-Seq analysis (Figure 5A–J). High variable expression patterns were observed among these genes. In both the ET and the control, six genes (ABC-B11, -C4, -C5, -C8, -G22, and -PDR1) showed a remarkable upregulation at 3000 GDH, while the other four genes showed a general decline from endodormancy to ecodormancy. The expression of ABC-C4 and -C8 in the ET treatment was lower at 3000 GDH compared to the control. It is worth noting also that the expression of ABC-PDR2 was significantly higher in ET-treated samples at 200 CH than in the control, reflecting its potential involvement in endodormancy establishment.

### 2.4. Annotation of DEGs between ET Treatment and Control during Ecodormancy

At 1000 GDH, upregulated DEGs were enriched in four categories, with three related to antioxidant activity (Figure 6A) and one in the cofactor-binding cluster. The downregulated DEGs yielded two GO terms, one related to the hydrolysis of glycosyl compounds and the other related to lipid metabolism. At 3000 GDH, the DEGs upregulated in ET were enriched in only 1 GO term of protein dimerization, whereas the downregulated DEGs generated up to 36 terms. Among these, nine were related to amino-sugar metabolism, three related to transmembrane transport, and four related to cell wall modification, mainly composed of genes encoding pectinesterase or polygalacturonase. Notably, this group also yielded four terms related to the defense response, in which three are associated with defense against fungal infections through chitin metabolic processes.

### 2.5. KEGG Pathway Analysis Revealed Additional Biological Functions from the DEGs

To gain further insights into the ET-responsive pathways, the KEGG enrichment analysis was performed on the DEGs between the ET treatment and the control at each time point. With the exception of the 1000 GDH group, only the downregulated DEGs from each sampling time were significantly enriched (Figure 7). The DEGs from 200 CH were enriched for only one pathway, the MAPK signaling pathway. The DEGs from the 600 CH were enriched in three pathways, all of which were related to DNA replication processes. At 1000 CH, most DEGs were enriched in chromosome-associated proteins, followed by glycolysis and gluconeogenesis, and a handful of DEGs in cyanamino acid metabolism and antenna proteins. All upregulated DEGs from 1000 GDH were enriched for DNA replication processes. The downregulated DEGs from 3000 GDH were enriched for a diverse range of pathways, which were mostly related to the release of ecodormancy.

### 2.6. Expression Profiles of Genes That Are Closely Related to the Biological Processes Revealed by the Enrichment Analyses

The GO and KEGG enrichment analyses revealed that DNA replication and transporter activity are two major events that were associated with the ET treatment during endodormancy. To gain insights into the functions of these key processes, we extracted the expression of four sets of genes encoding enzymes that are closely associated with these processes. Among these, cyclins are responsible for the cell cycle; aquaporins control the movement of water through cellular membranes; and β-1,3-glucan synthase and β-1,3-glucanase mediate callose accretion and degradation, respectively. Each gene set comprised all of the putative homologs for each categorical gene from the transcriptome, and their general and individual expression profiles were presented as average z-scores and heatmaps, respectively (Figure 8).

The 16 retrieved cyclin genes include 8 cyclin D, 3 cyclin A, and 2 cyclin B genes. According to their expression patterns, these genes were evenly divided into two groups, with one group upregulated in endodormancy and the other group upregulated in ecodormancy (Figure 8A). During endodormancy, the expressions of four genes (Prupe.4G034400, Prupe.8G137700, Prupe.8G177900, and Prupe.4G269600) were lower in the ET treatment than the control at 600 and 1000 CH. Four groups of aquaporin intrinsic protein (IP) genes were extracted in this study, including plasma membrane (PIP), tonoplast (TIP), small basic (SIP), and nodulin-26-like (NIP) (Figure 8B). Most of these genes were highly transcribed during ecodormancy, with six upregulated at 1000 GDH and the other six at 3000 GDH. Noticeably, three genes (P.3G063500, P.1G449500, and P.7G000900) showed downregulation in the ET treatment at 3000 GDH. The nine β-1,3-glucan synthase genes exhibited distinct expression patterns between endodormancy and ecodormancy, with upregulation in the former and downregulation in the latter (Figure 8C), strongly suggesting active callose synthesis during endodormancy. At 1000 CH, most genes showed either a slight or marked upregulation in the ET treatment. In the search for genes encoding β-1,3-glucanase, we identified 42 putative homologs, 5 of which were chosen for presentation as they were recently identified within a QTL related to bud dormancy [37]. In these five genes (Figure 8D), two were upregulated during endodormancy, and the other three upregulated in ecodormancy, indicating their distinct functions at different stages of dormancy.

### 2.7. High Similarity between Transcriptome Profiles and RT-qPCR Based Gene Expression of DAM Genes

As highly important regulators of dormancy, the DAM genes consist of six tandem duplication genes in peach. In this study, the expression levels of DAM 2 and 3 were extremely low and excluded from presentation. Based on the transcriptome profile, four DAM genes, DAM 1, 4, 5, and 6, were highly expressed at 200 CH and then declined gradually until the end of ecodormancy (Figure 9). No significant difference was found between the ET-treated and control samples at any time point. Notably, DAM 5 and 6 showed nearly identical expression patterns, both reaching the lowest point at 1000 CH, and remained low until the end of ecodormancy. In the qPCR validation of DAM gene expression, highly similar expression profiles were found among all four DAM genes, which bore high similarity with DAM 5 and 6 as revealed in the RNA-Seq analysis.

## 3. Discussion

Bud dormancy is a critical stage in the life cycle of tree fruits, directly influencing flowering, fruit set, and the eventual fruit yield. Over the last several decades, the phenology of dormancy and flowering in many perennial species has experienced an unprecedented disturbance from global climate change, which raised growing concerns in the sustainability of fruit production. Recent years have seen increasing research on the mechanisms underpinning dormancy in various perennial species using omics-based approaches. In general, these studies lack the inclusion of treatments that have direct effects on the phenology of dormancy and flowering. Owing to its manifested effects in extending CR and delaying the bloom time in peach (Figure 1A), ET provides an elegant model to decipher the dormancy mechanism. In this study, RNA-Seq analysis revealed the distinct transcriptome profiles of floral buds between the ET-treated and control samples during endodormancy and ecodormancy (Figure 2 and Figure 3), which reflect the dynamics of the underlying biological processes.

Throughout the entire dormancy period, the peach floral buds undergo continuous development in which cell division, differentiation, and enlargement take place [11]. In this study, the GO enrichment analysis revealed most of the downregulated DEGs at 600 CH were enriched in processes associated with growth (Figure 4), including DNA replication, saccharide metabolism, and the hydrolysis of ATP. In particular, four downregulated DEGs encode kinesins, a type of motor protein that regulates both cell expansion and cell division, and they are also responsible for the transport of various cellular components [38]. Consistently, the KEGG enrichment analysis of these DEGs yielded three pathways that were all associated with DNA replication (Figure 4). Clearly, these results suggest that ET halted the cellular processes and the development of floral buds during endodormancy.

Cell division is dictated by the cell cycle, which is an essential aspect of bud dormancy [39,40]. The progression of the cell cycle is primarily governed by the activity of cyclin-dependent kinases (CDKs), in partnership with cyclins, in which type A, B, and D cyclins are in the main regulators, controlling critical transitions in the cell cycle [38]. In grapevine (V. vinifera) buds, Vergara et al. [41] indicated that the depth of dormancy is closely correlated with the expression levels of cyclins, which can be downregulated by hydrogen cyanamide, a dormancy-breaking compound. In this study, two cyclin genes at 600 CH (*Cyclin-B2-3* and *Cyclin-D3-3*) and two at 1000 CH (*Cyclin-B1-2* and *Cyclin-U4-2*) were downregulated in the ET treatment (Figure 4). Examination of the 16 retrieved cyclin genes in peach indicated that cell cycle processes in endodormancy and ecodormancy are controlled by two different groups of cyclin genes (Figure 8A), with the four abovementioned genes more involved in ecodormancy, and the rest more active in endodormancy. The involvement of cyclins during endodormancy was also demonstrated in a transcriptome study on apricot buds, in which the expression of cell cycle genes, including three Cyclin-D genes and one cell cycle checkpoint control protein, are upregulated during endodormancy [12], indicative of active cell division at this stage. Therefore, the reduced expression of cyclin genes in ET treatment at 600 and 1000 CH (Figure 8A) indicated that ET repressed the cell cycle, which may hinder the development of floral organs during endodormancy.

ATP-binding cassette (ABC) transporters are a large family of transmembrane proteins that conduct across-membrane translocation of a wide range of substrates. The classification of ABC transporters is based on the configuration of the essential domains, which confers the specificity of the substrates that they transport [42]. Many ABC transporters play important roles in detoxification and pathogen defense through the active transport of secondary metabolites and xenobiotic compounds [43]. In this study, 10 ABC transporter genes were differentially expressed in response to the ET treatment, with their overall expression peak shifted from 600 CH in the control to 1000 CH in the ET-treated samples (Figure 5). The qPCR validation of these genes illustrated such a shift could be driven by the upregulation of *ABCC4*, *ABCC5*, and *ABC-PDR1* at 1000 CH (Figure 5). The transporter ABCC4 plays a vital role in the detoxification process in both animals and plants, and it has also been implicated in other cellular processes, such as the regulation of stomatal aperture [44]. In soybean, the ABCC5 was identified as a transporter of phytate [45], a main phosphorus-containing compound that can increase osmotic tolerance through the stimulation of antioxidant systems [46]. The ABC proteins in the pleiotropic drug resistance (PDR) family have unique domain organizations and are implicated in responses to abiotic and biotic stress, in the latter case, by pumping antimicrobial compounds out of the cell [47]. The upregulation of three ABC transporter genes in this study therefore indicated that ET may induce elevated stress responses during endodormancy. This hypothesis can also be supported by the observation that ET stimulates the production of gummosis (Figure 1B), which is a typical symptom of stress in stone fruits [1]. Indeed, pathways governing stress response and bud dormancy have been proposed to converge in peach through shared regulators, such as gene *PpSAP1*, which activates responses common to the two processes [48]. Therefore, this ET-induced stress response may in turn intensify dormancy and delay the release of endodormancy and the eventual flowering time.

In addition to conferring physical characteristics, cell walls also undergo remolding and modification to accommodate bud growth and development during dormancy. Under a short-day photoperiod, the permeability of bud cell walls decreases through enzymatic modification, thereby facilitating the establishment of dormancy [49]. In Persian walnut (*Juglans regia* L.), Gholizadeh et al. [50] observed that budbreak is accompanied by enhanced activities of hemicellulase and pectinase, which respectively hydrolyze cellulose and pectin in the scale parenchyma cells. The cellulose and pectin are essential in maintaining the integrity of the cell wall, and their degradation leads to a loosened cell wall, increasing the cell wall permeability and allowing for the outgrowth of bud primordia. This degradation also releases the carbohydrate reserves, providing the nutrients necessary for the budbreak. In this study, ET downregulated many genes that are responsible for cell wall degradation at 1000 CH and 3000 GDH (Figure 6 and Figure 7), which may reduce the activities of these processes and thereby delay the alleviation of endodormancy and ecodormancy. The essential role of cell wall modification in budbreak was confirmed in a recent transcriptome study by Zhao et al., 2020, which demonstrated that cell wall modification is controlled by the transcription factor EARLY BUDBREAK 1 (EBB1), a positive regulator of budbreak. In grapevine, Sudawan et al. [19] indicated that cell wall loosening can be induced through increased levels of ROS prior to budbreak. Hence, the relaxation of cell walls and the accompanied polysaccharide mobilization are an essential aspect of bud growth and development. In this study, ET-mediated reduction in these processes may contribute to extended ecodormancy and the subsequent delayed bloom.

Accumulating evidence has pointed to the central role of reactive oxygen species in controlling dormancy progression [16]. The dynamics of ROS have extensive influence over regulatory mechanisms, such as hormonal signaling, modification of the plasma membrane, carbohydrate dynamics, mitochondrial respiration, and oxidative stress. In this study, the expression profiles of upregulated DEGs at 1000 GDH revealed a distinguishable shift from 1000 CH to 1000 GDH due to the ET treatment. In these DEGs, three GO terms were related to the antioxidant system, including peroxidase activity, antioxidant activity, and oxidoreductase activity (Figure 4). Previous studies indicate that the accretion of ROS, coupled with antioxidant scavenging mechanisms, is essential for the release of endodormancy and ecodormancy [16,19]. Consistently, our recent study [3] showed the ET treatment induces higher levels of ROS (e.g., singlet oxygen and hydrogen peroxide) and the elevated activity of ROS-generating enzymes (e.g., NADPH-oxidase and superoxide dismutase), as well as scavenging enzymes (e.g., catalase and glutathione peroxidase) during endodormancy. In this study, the ET-induced expression shift of genes that are related to antioxidant activities confirmed the above findings and also signified the effects of ET in modulating ROS levels, which altered the phenology of dormancy and flowering.

The development of dormant buds also heavily relies on the mobility of water. At the molecular level, water flow is controlled by water channels, known as aquaporins, the transmembrane proteins gating the intracellular and intercellular transport of water, as well as some other small solutes [51]. The two most abundant aquaporins are TIP and PIP, which are integral to vacuolar and plasma membranes, respectively [52]. In a transgenic study, TIP genes in plum (*P. domestica*) buds were found to be suppressed by the overexpression of *PpSAP1*, a gene encoding stress-associated protein, that has dual roles in regulating both stress responses and dormancy [53]. Further, the expression of two aquaporin genes, deltaTIP and PIP2, in peach buds was shown to increase upon the release of endodormancy, accompanied by the increase in water content in the primordia and adjacent tissues [54], indicating the increased demand of water after buds entering ecodormancy. Similarly, this study showed 12 out of the 14 aquaporin genes were highly expressed in ecodormancy in a time-specific pattern, with 6 genes expressed at 1000 GDH and the other 6 at 3000 GDH (Figure 8B), suggesting their distinct functionality at different stages of ecodormancy. The expression of four genes were lower in the ET treatment compared to the control at 3000 GDH. The downregulation of these aquaporin genes was in line with our finding in PD permeability, both reflecting a reduction in transmembrane transport in the ET treatment. The findings of this study support the notion that aquaporins are important in facilitating water movement and regulate the acquisition of the developmental competency of dormant buds, thus controlling dormancy progression.

In plants, the intercellular transport in the symplastic space relies on membranous channels between adjacent cells known as plasmodesmata (PD). The connectivity of PD is controlled through the deposition and removal of callose, which are catalyzed by enzymes of β-1,3-glucan synthase and β-1,3-glucanase, respectively [55]. PD-mediated symplastic closure is a key mechanism in response to pathogens or abiotic stresses and also a critical step in the dormancy establishment [56,57]. In this study, nine β-1,3-glucan synthase genes were upregulated during endodormancy, while downregulated in ecodormancy in both the ET treatment and the control (Figure 8C), suggesting callose deposition is exclusively specific to endodormancy. At 1000 CH, the majority of the β-1,3-glucan synthase genes in the ET treatment were slightly or markedly upregulated relative to the control samples, which may contribute to the intensification and extension of endodormancy. Of the 42 retrieved β-1,3-glucanase homologs (Figure 8D), which are responsible for callose degradation, 5 were candidate genes in the previously identified QTL involved in bud dormancy [37]. These genes showed time-specific expression characteristics, each upregulated at only one time point, indicating these genes have distinct functions that are specific to dormancy stage. A previous study indicated that deposition of callose at the PD during dormancy also prevents the passage of growth factors, such as FT into the buds, thereby inhibiting floral organ development and delaying dormancy release [58]. Further, the permeability of PD is under direct control of ABA, a central regulator of dormancy, confirming the involvement of PD in the progression of dormancy [21]. Our result confirmed the important role of PD in dormancy and revealed that ET induced high callose synthesis, which may likely intensify endodormancy as well as inhibit floral development.

DAM genes play critical roles in regulating dormancy initiation and maintenance in stone fruits. In peach, six DAM genes are arranged in tandem at the EVG locus and have highly similar sequences [59]. *DAM1*, *4*, *5*, and 6 are responsive to a short photoperiod and chilling accumulation and are believed to be responsible for dormancy control [31]. In this study, the RNA-Seq data successfully profiled these four DAM genes, which all exhibited high expression levels at the early stage of endodormancy, followed by a gradual decline as dormancy progresses (Figure 9). The remarkably higher expressions of *DAM5* and *6*, compared to *DAM1* and *4*, are consistent with the finding that *PpDAM5* and 6 are the stronger candidate genes in the QTL controlling CR and the bloom time in peach [60]. Importantly, the RNA-Seq and subsequent validation by qRT-PCR indicated that there was no differential expression in DAM genes between the control and the ET treatment. Similarly, Tang et al. [61] reported that the *DAM1*, *5*, and *6* transcript levels in peach were not affected by the application of hydrogen cyanamide, a widely used budbreak agent. As an essential regulatory hub in dormancy, the robust expression profile of DAM genes would be necessary to buffer changes from environmental fluctuations, as such to intrinsically regulate the initiation and progression of dormancy. Relatively, their roles in dormancy release, floral development, and budburst are probably not as critical, as the avg peach mutant where DAM genes are lacking can still flower normally in permissive conditions [62].

## 4. Materials and Methods

### 4.1. Plant Materials and Treatments

The peach cultivar used in this study was 7-year-old ‘Redhaven’ grown at the Alson H. Smith JR Agricultural Research and Extension Center (AREC), Winchester, VA, United States of America (39.11, −78.28). Six ‘Redhaven’ trees of similar size were selected for this experiment, in which three received ethephon treatment and the other three used as untreated controls. At least five buffer trees were assigned between ET-treated and control trees. At 50% leaf fall (24 October 2019), ethephon (Motivate, Fine American Inc., Walnut Creek, CA, USA) at 500 ppm, mixed with a non-ionic surfactant, Regulaid (Kalo Inc., Overland Park, KS, USA), at 250 ppm was applied using a high-pressure tree sprayer. Field temperatures were monitored at 10 min intervals using a Temperature Data Logger (EL-USB-1, Contoocook, NH, USA.). Chilling accumulation in the field was calculated as chilling hours (CH): the number of hours with temperature in the range of 0–7.2 °C during dormancy period [6]. Peach buds were sampled at eight time points based on the accumulation of chilling hours (20, 200, 400, 600, 800, and 1000 CH) or growth-degree hours (1000 and 3000 GDH), which corresponded to calendar dates of 14 October, 11 and 27 November, 9 and 27 December in 2019, 15 January, 14 February, and 12 March in 2020, respectively. At sampling, about 20 floral buds were collected from each tree and pooled as a biological replicate. Sampled buds were immediately frozen in liquid nitrogen, then stored at −80 °C until further process. CR and HR were determined by evaluating budbreak of branch cuttings after being kept in permissive conditions as detailed by Liu et al. [2].

### 4.2. RNA Extraction and Sequencing

Total RNA was extracted from 1 g of ground bud tissue according to a CTAB method [63], as it produces high RNA yield, purity, and integrity from plant samples. All RNA extracts were purified using an RNA purification kit (Zymo Research, Irvine, CA, USA). The RNA sequencing was performed at Novogene (El Monte, CA, USA) using an Illumina NovaSeq platform with a paired-end 150 bp sequencing strategy. The sequencing was performed for 30 bud samples comprising three time points during endodormancy (200, 600, and 1000 CH), two time points during ecodormancy (1000 and 3000 GDH), two treatments (ET-treated and control trees), and three biological replicates per treatment.

### 4.3. Library Construction, Quality Control, and Reads Alignment

After the QC procedures, all mRNA samples were enriched using oligo(T) beads. First, the mRNA was fragmented randomly by adding fragmentation buffer, and then the cDNA was synthesized based on mRNA template and random hexamers primer. Next, a custom second-strand synthesis buffer (Illumina), dNTPs, RNase H, and DNA polymerase I were added to initiate the second-strand synthesis. Then, after a series of terminal repair a ligation, and sequencing adaptor ligation, the double-stranded cDNA library is completed through size selection and PCR enrichment.

RNA-Seq produced an average of 24 million reads per library. Raw reads in fastq format were first processed using Novogene’s in-house Perl-based program, which removes reads containing adapter or ploy-N (N > 10%, N represents undetermined bases), or reads of low qualities (Qscore ≤ 5). Processed reads were subsequently subjected to fastp (Shen et al., 2018) to filter out rRNA when the rRNA rate > 15%. All the downstream analyses were based on the cleaned reads with an average clean read rate of 97.2%. The cleaned reads were aligned using program HISAT2 [64] against the peach reference genome at https://www.rosaceae.org/species/prunus/persicav1.0 accessed on 26 May 2022. A total of 28,798 genes were successfully mapped, with an average mapped rate of 95.5% and uniquely mapped rate of 93.6%. To remove technical biases and facilitate between sample comparison, raw counts were normalized into FPKM (fragments per kilobase of transcript per million reads mapped) FPKM.

### 4.4. PCA and Cluster Analysis of Gene Expression

The overall similarity was assessed using principal component analysis (PCA) in which the FPKM expression values were used for all the 30 samples. To visualize relationships between the ET treatment and the control at each time point, a hierarchical clustering analysis was conducted on the normalized log_2_(FPKM + 1) of the mean expression values over three biological replicates. In this analysis, genes showing similar trends in expression levels under different conditions were clustered together.

### 4.5. Differential Gene Expression Analysis and Functional Analysis

Differential analysis was performed using the DESeq2 R package [65] to examine the expression-level differences between the ET treatment and the control. The differentially expressed genes (DEGs) were determined using the threshold of |fold change (FC)| > 2 (or |log2FC| > 1) and adjusted *p*-value (Padj) < 0.05, which was adjusted using the Benjamini–Hochberg Procedure to control the false discovery rate (FDR).

To determine which biological functions or pathways the DEGs of interest are significantly associated with, we performed the GO (Gene Ontology) enrichment and KEGG (Kyoto Encyclopedia of Genes and Genomes) pathway enrichment analyses using the clusterProfiler [66] software. In these analyses, a hypergeometric distribution model was used to test whether genes from GO or KEGG functional terms are enriched in the set of DEGs to be tested. The *p*-value was adjusted by the Benjamini-Hochberg Procedure, and GO terms or KEGG pathways with a P-adj < 0.05 were considered as being significant. The enrichment analyses were conducted on the DEGs from the comparison groups of endodormancy vs. ecodormancy; ET treatment vs. control across all the time points; and ET treatment vs. control at each time point. To illustrate the expression pattern of the DEGs from the groups of ET treatment vs. control at each time point, the mean expression level in FPKM was calculated as the average z-scores of the genes within each group: *z-score* = (*FPKMij − meani*)/*sdi*, where *FPKMij* is the FPKM value of the *ith* gene in the *jth* sample, *meani* and *sdi* are the mean and standard deviation of the *ith* gene across all samples. To complement the annotation of key processes identified by GO and KEGG enrichment analyses, we extracted the expression of four gene sets that were closely related to these processes (Appendix A). The selection was performed by searching for all the homologs of a target gene in the transcriptome and the mean expression of the given gene set was represented using the average z-score.

### 4.6. RT-qPCR Validation

Genes that were characterized from the transcriptomic analysis were validated by gene expression analysis using quantitative real-time PCR (qPCR). In this analysis, RNA was first reverse transcribed to cDNA using a cDNA Synthesis Kit (Applied Biosystem, Foster City, CA, USA), then followed by qPCR reactions on a CFX Connect Real-Time System (Bio-Rad, Hercules, CA, USA). Each qPCR reaction was performed on three biological replicates in three technical replicates, using SsoFast EvaGreen Supermix (BioRad, Hercules, CA) as the dsDNA-binding dye. The primers were designed using Primer3Plus (https://www.bioinformatics.nl/cgi-bin/primer3plus/primer3plus.cgi, accessed on 24 March 2020) for target genes. Two peach housekeeping genes, ß-actin and ubiquitin, were used as internal controls. The normalized relative expression for each gene was statistically analyzed using the CFX manager software (Bio-Rad). The primers used in the validation analyses were provided in Appendix A. Gene expression profiling was conducted on all bud samples collected throughout the dormancy cycle, including those that were not analyzed by RNA-Seq.

## 5. Conclusions

The transcriptome analyses in this study revealed that ET induced stress responses during endodormancy and delayed several biological processes related to cell division and intercellular transport, which are essential for the development of floral organs. During ecodormancy, ET suppressed pathways related to antioxidant processes and cell wall modification, both of which are closely associated with budburst and flowering. Interestingly, the expression of dormancy-associated MADS (DAM) genes remained relatively unaffected by ET, consistent with their conserved nature. The findings of this study signified the importance of floral organogenesis during dormancy, in which several processes are subject to the influence of ET and may serve as possible targets for other PGRs in mitigating frost damage.

## Figures and Tables

**Figure 1 ijms-23-06801-f001:**
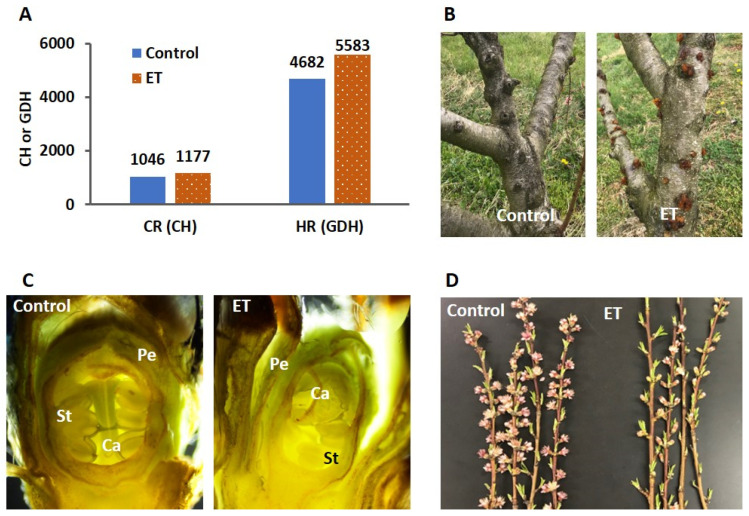
Phenotypical characterization of the ET treatment on dormant peach buds. (**A**) Chilling requirement (CR) and heat requirement (HR) of the control and the ET treatment. (**B**) Comparison of gummosis incidence on the tree barks of the ET treatment and the control. (**C**) Dissection of dormant floral buds from the control and the ET treatment at 1100 CH, showing petal (Pe), stamen (St), and carpel (Ca). (**D**) Bloom of the control and the ET treatment on 30 March 2020. CH, chilling hour; GDH, growing degree hour; ET, ethephon.

**Figure 2 ijms-23-06801-f002:**
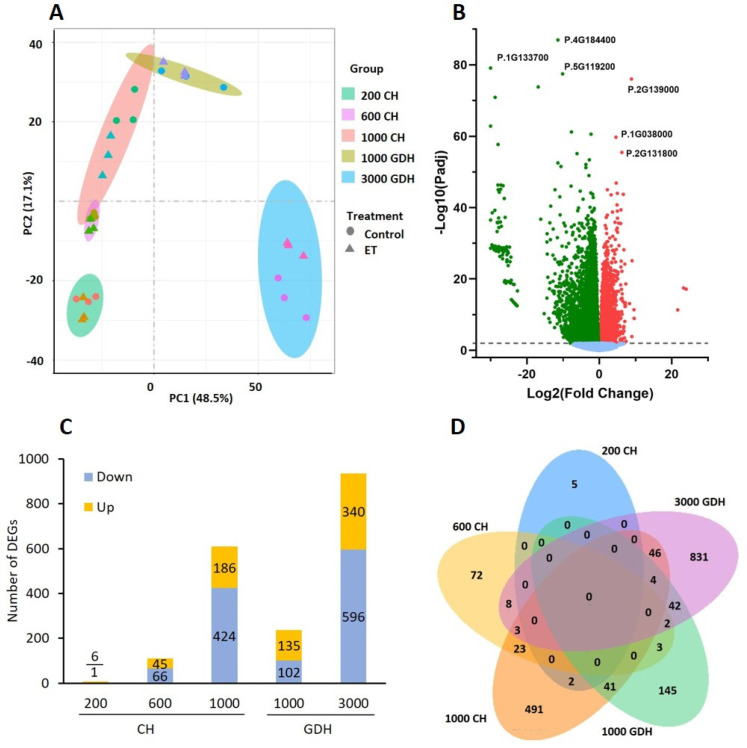
Transcriptome profiles of peach buds in the ET treatment and the control during endodormancy and ecodormancy. (**A**) Principal component analysis (PCA) of transcriptomic profile of the ET treatment and the control in floral buds of peach. (**B**) Volcano plot of 11,582 non−DEGs (light blue) and 15,053 DEGs (ecodormancy vs. endodormancy), with 7089 upregulated (red) and 7964 downregulated (green). (**C**) The numbers of DEGs (|Log2 fold change| > 1, FDR < 0.05) arising from the ET treatment at five sampling times (CH: chilling hour; GDH, growth-degree hour). (**D**) Venn diagram of DEG distribution at each sampling time.

**Figure 3 ijms-23-06801-f003:**
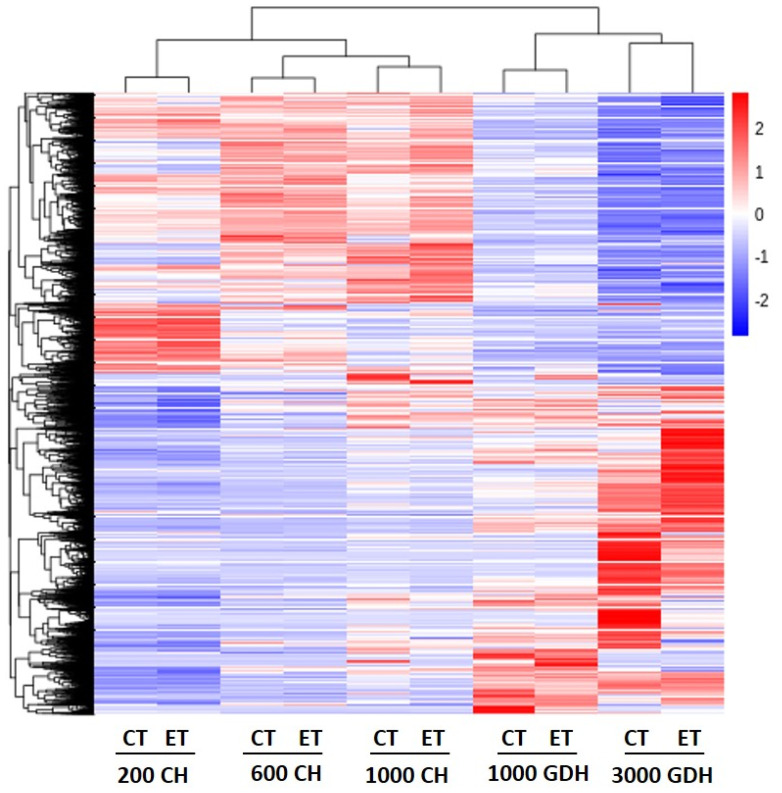
Hierarchical clustering analysis of all peach genes. Each datapoint consists of the expressions (FPKM) of three biological replicates. Red indicates high relative gene expression and blue indicates low relative gene expression. CT, control; ET, ethephon treatment.

**Figure 4 ijms-23-06801-f004:**
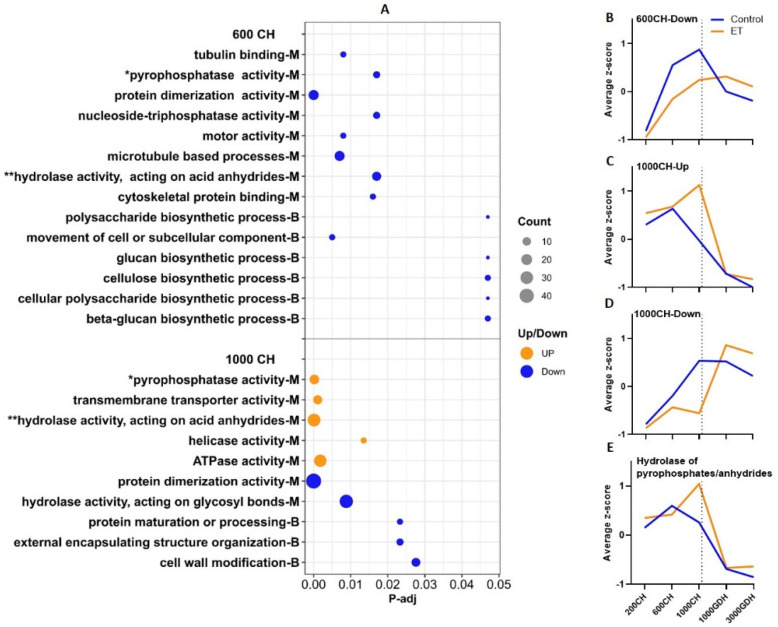
Enrichment analysis of DEGs between the ET treatment and the control during endodormancy (200, 600, and 1000 CH). (**A**) Significantly enriched GO terms (P−adj < 0.05). Letter B indicates biological processes, and M indicates molecular functions. Dot size represents the number of genes enriched in each GO term. Orange dots represent upregulated DEGs, and blue dots represent downregulated DEGs, respectively. Asterisks (* and **) indicate GO terms that are common to both 600 CH and 1000 CH sampling time. (**B**–**E**) Mean z−score values of the expression (FPKM) of the DEGs from each group. The dotted lines represent the date of endodormancy release.

**Figure 5 ijms-23-06801-f005:**
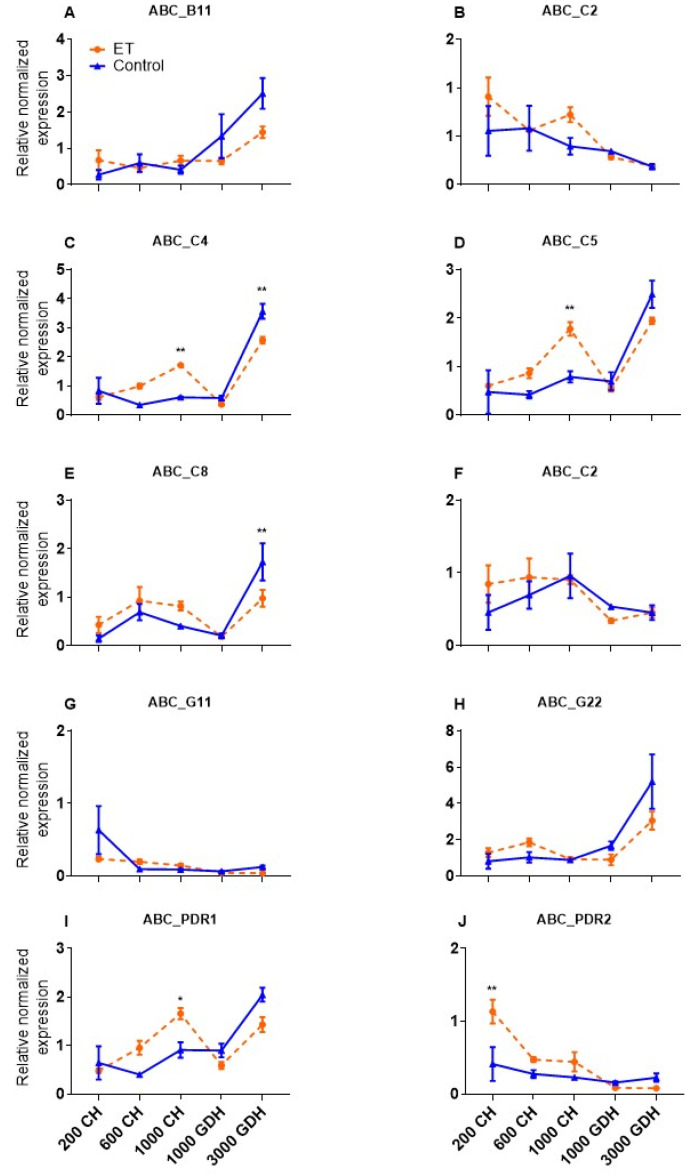
RT-qPCR validation of the expression of 10 ABC transporter genes (**A**–**J**), which were downregulated at 600 CH and upregulated at 1000. Each datapoint represents the mean ± SE of three biological replicates, each with two technical replicates. Significant differences are marked by asterisks (* *p* < 0.05, ** *p* < 0.01).

**Figure 6 ijms-23-06801-f006:**
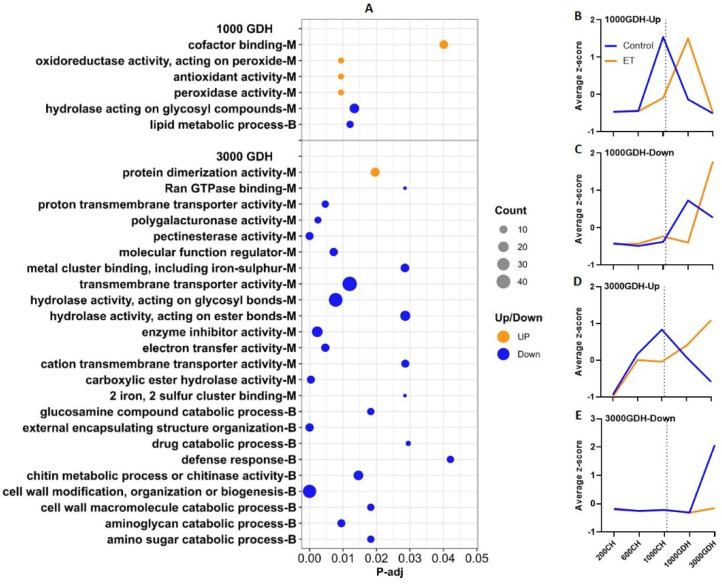
Enrichment analysis of DEGs between the ET treatment and the control of peach cultivar ‘Redhaven’ during endodormancy (1000 and 3000 GDH). Refer to Figure 4 for detailed legend description. (**A**) Significantly enriched GO terms (P-adj < 0.05). Letter B indicates biological processes, and M indicates molecular functions. Dot size represents the number of genes enriched in each GO term. Orange dots represent upregulated DEGs, and blue dots represent downregulated DEGs, respectively. (**B**−**E**) Mean z-score values of the expression (FPKM) of the DEGs from each group. The dotted lines represent the date of endodormancy release. The expression profiling showed a prominent peak shift in the 1000 GDH upregulation group (Figure 6B), in which the ET treatment and the control yielded nearly identical peaks, with the ET treatment peaking at 1000 GDH, lagging behind the control, which peaked at 1000 CH (Figure 6C), consistent with ET retarding the expression of genes in this group. Similarly, in the 1000 GDH downregulation group, the control peaked at 1000 GDH, while the ET treatment reached a maximum at 3000 GDH. Opposite expression patterns were observed in the 3000 GDH upregulation and downregulation groups, wherein the control peak appeared at 1000 CH, while the ET treatment reached the highest level at 3000 GDH. In the 3000 GDH downregulation group, the control was nearly identical to the ET treatment, but the latter underwent a drastic increase at 3000 GDH (Figure 6D) which was not observed in the control (Figure 6E).

**Figure 7 ijms-23-06801-f007:**
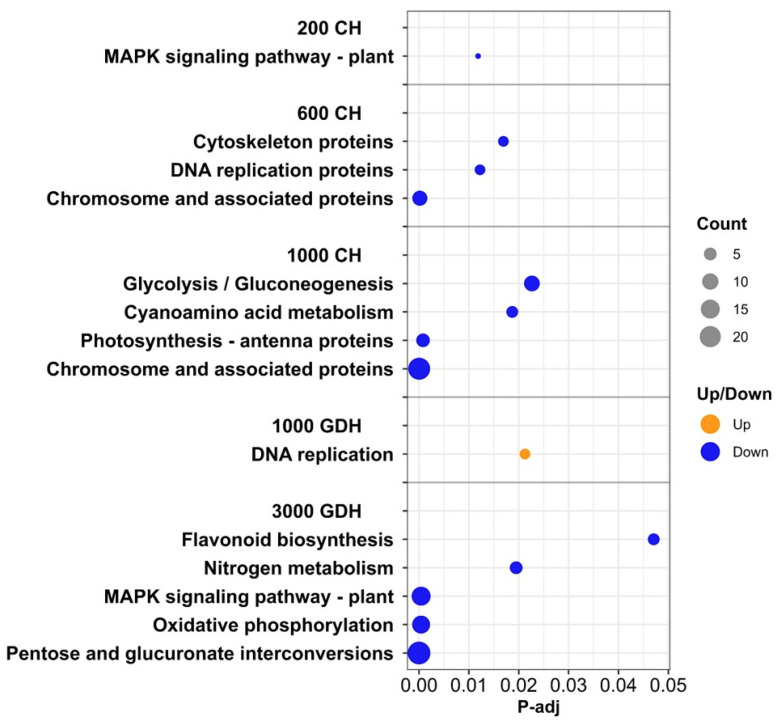
Significantly enriched KEGG pathways of the DEGs between the ET treatment and the control from five time points during endodormancy and ecodormancy. Dot size represents the number of genes enriched in each KEGG pathway. Orange dots represent up-regulated DEGs, and blue dots represent downregulated DEGs, respectively.

**Figure 8 ijms-23-06801-f008:**
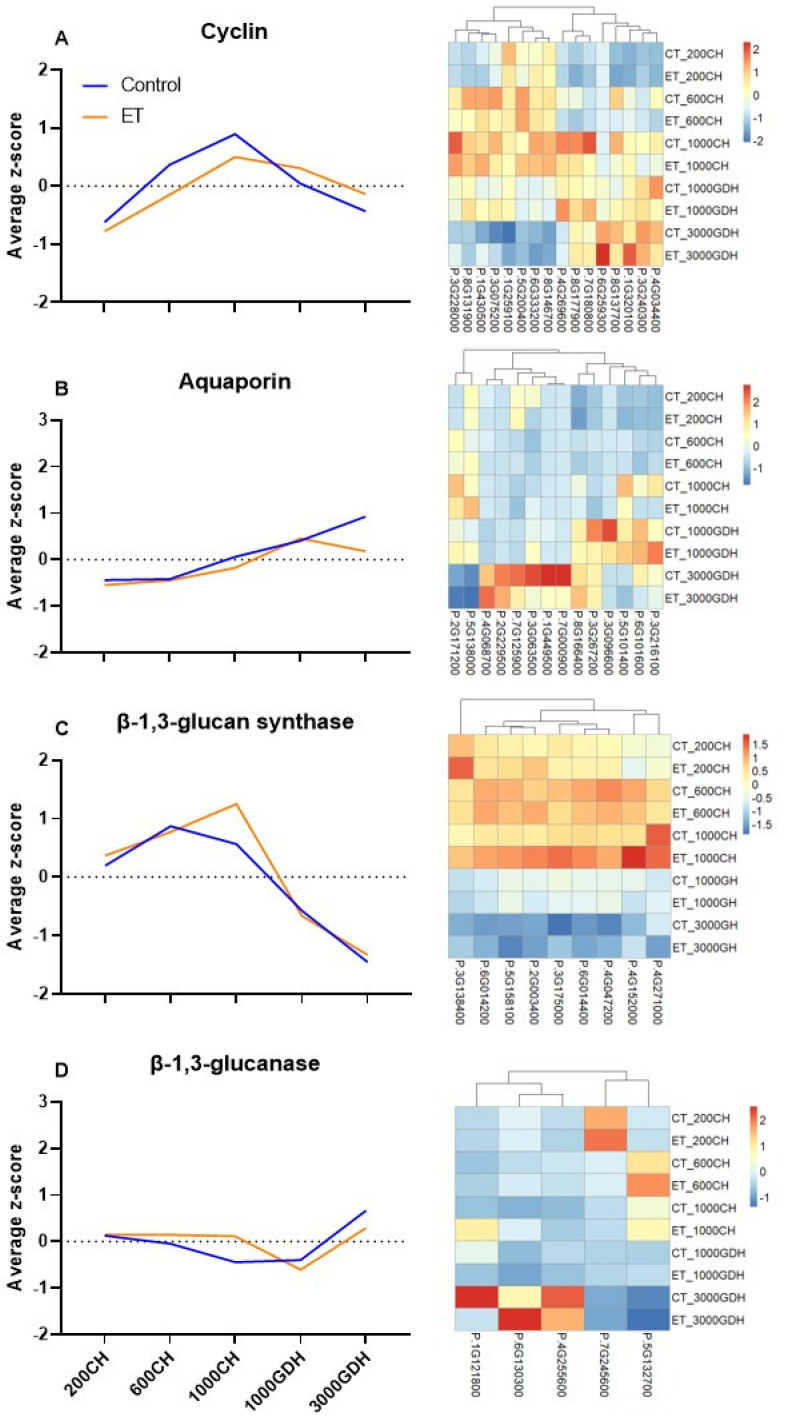
Expression patterns of four genes that are closely associated with cell division and intercellular transport. (**A**−**D**) mean z-score values (on the left) and heatmap (on the right) of normalized FPKM values of each gene.

**Figure 9 ijms-23-06801-f009:**
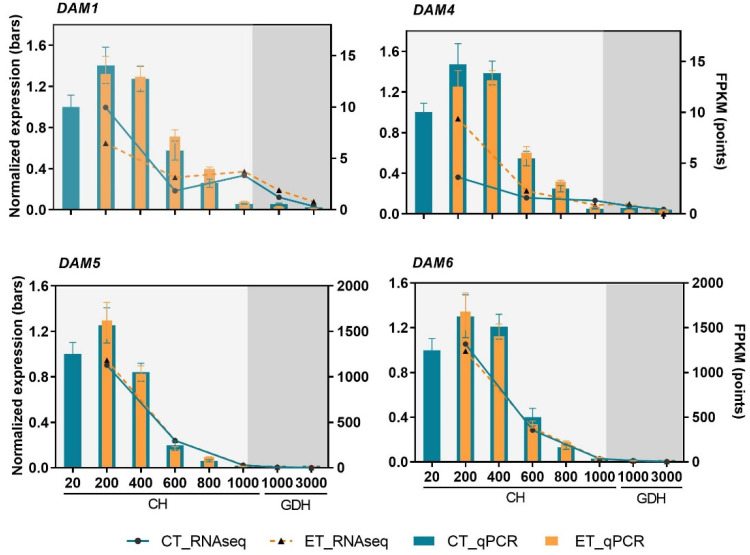
Expression profiles of DAM genes represented as FPKM values (points) and validation using RT-qPCR expression (bars), in which each datapoint represents the mean ± SE of three biological replicates, each with two technical replicates.

## Data Availability

The data presented in this study are available in the Appendix A.

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
