# Peer review of "Fall Applications of Ethephon Modulates Gene Networks Controlling Bud Development during Dormancy in Peach (Prunus Persica)"

_ijms, 2022, doi:10.3390/ijms23126801_

Round 1

Reviewer 1 Report

Liu et al. have presented results on Modulating the gene networks that controlling bud development during dormancy in peach by applications of ethephon. The topic is relevant from both scientific and practical point of view and suitable for the scope of the IJMS. However, the manuscript contains some errors that are mentioned below.

Firstly, the authors should consider the RNA isolation methods. They indicated that they have used CTAB method for RNA extraction but to ensure a successful RNA-Seq experiment, the RNA should be of sufficient quality to produce a library for sequencing and obtaining RNA of the highest purity is the best assurance that the downstream manipulations will be successful. So, please mention why you did you used CTAB method particularly? or why did you prefer it rather using an accurate commercial kit?

Secondly, some figures need to be revised as:

Figure 2C and 2D: the legend description is wrong for both “(C) is not Venn diagram of DEG distribution at each sampling time”, please revise and replace.

Figure 2C: “The numbers of DEGs arising from the ET treatment at five sampling time (CH: chilling hour; GDH, growth degree hour)”. You have mentioned that you did FDR to examine the significance of the DEG, you should put a significance symbol on the figure and mention to this in the legend.

Figure 2D: the resolution of the Venn diagram is not good, please improve it.

Figure 9. you have mentioned in the legend that the significant differences are marked by asterisks (* P < 0.05, ** P < 0.01), please add asterisks.

Author Response

Thank you for reviewing the manuscript and providing valuable comments. As per your suggestions, revisions or modifications have been made accordingly.

Firstly, the authors should consider the RNA isolation methods. They indicated that they have used CTAB method for RNA extraction but to ensure a successful RNA-Seq experiment, the RNA should be of sufficient quality to produce a library for sequencing and obtaining RNA of the highest purity is the best assurance that the downstream manipulations will be successful. So, please mention why you did you used CTAB method particularly? or why did you prefer it rather using an accurate commercial kit?

We've been using this modified CTAB method routinely in our lab, and are successful in obtain high quality RNA each time. The company which performed the RNA-Seq assured us of using this method before we submitted our samples. 

Brief justification was provided for using CTAB in RNA extraction. Line: 582

Figure 2C and 2D: the legend description is wrong for both “(C) is not Venn diagram of DEG distribution at each sampling time”, please revise and replace.

Legend description was corrected. Line: 347-349

Figure 2C: “The numbers of DEGs arising from the ET treatment at five sampling time (CH: chilling hour; GDH, growth degree hour)”. You have mentioned that you did FDR to examine the significance of the DEG, you should put a significance symbol on the figure and mention to this in the legend.

FDR cutoff and fold change threshold for generating DEGs was added to the legend. Line: 347

Figure 2D: the resolution of the Venn diagram is not good, please improve it.

The Venn diagram was redone, and the resolution was improved. 

Figure 9. you have mentioned in the legend that the significant differences are marked by asterisks (* P < 0.05, ** P < 0.01), please add asterisks.

There were no significant differences in this graph. The unnecessary description was removed. 

Reviewer 2 Report

IJMS Review Report - -

General comments:

In general, the manuscript titled has a valuable topic. the work had a significant contribution to the field. The manuscript is well written. the work scientifically sound and not misleading The English language and style are fine and readable except for some moderate English language check required.

Are there appropriate and adequate references to related and previous work

 There are some MINOR comments.

Detailed comments:

In general, please avoid using the personal pronouns (I, We, our) as it was found line 21 (our findings), and more. Please apply this rule throughout the manuscript.

Title:

It ok

Keywords:

The keywords list was carefully and accurately chosen

Abstract:

The aim of the study and the main objectives were clearly stated.

Please add some values and significant findings to this section.,

Introduction:

This section didn’t provide enough background about the topic. The introduction needs to be elongated and enriched.

Materials and Methods:

The experimental design is adequate and suitable to the current study.

Results:

The provided data is very interesting BUT some results were poorly presented.

Discussion:

This section is ok but some results were poorly discussed.

For example, Figure #3: hierarchical clustering Analysis: please discuss the data thoroughly

Also figure # 8 Mean Z-score and heatmap: Please explain and discuss the data thoroughly.

*In general, it was difficult to relate the discussion section to the corresponding data in the results section. Therefore, for best discussion to the provided data the author is strongly advised to combine the results section and the discussion section.

Conclusion:

This section is ok. This section provides a good conclusion for the study and includes the significant findings.

References:

The authors provided enough citations and it was UpToDate.

Author Response

Thank you for your reviewing and the helpful comments. The detailed modifications are listed as follows. 

Detailed comments:

In general, please avoid using the personal pronouns (I, We, our) as it was found line 21 (our findings), and more. Please apply this rule throughout the manuscript.

Some of personal pronouns were replaced. In this manuscript, the use of some personal pronouns is necessary for 2 reasons. First, it allows for the use of active voice, so the sentences are more direct, clear and concise, especially when the long subordinate clauses need to be put after the main clauses. Secondly, some the personal pronouns serve to link the present study to our previous investigations, so the readers can easily reference to them.

Abstract:

The aim of the study and the main objectives were clearly stated.

Please add some values and significant findings to this section.

Values and significance were added to the end of the Abstract. Line: 24-26

Introduction:

This section didn’t provide enough background about the topic. The introduction needs to be elongated and enriched.

Introduction was expanded to review dormancy related aspects such as: plant hormones, ROS, and intercellular communication. Line: 78-118

Results:

The provided data is very interesting BUT some results were poorly presented.

Discussion:

This section is ok but some results were poorly discussed.

For example, Figure #3: hierarchical clustering Analysis: please discuss the data thoroughly

Figure 3 was described and discussed in more details. Line: 119-205 and 401-404.

Also figure # 8 Mean Z-score and heatmap: Please explain and discuss the data thoroughly.

More details on Figure 8 were provided in Results. Line: 310-322

The Discussion was reorganized, and some details were added. The discussion of each panel in Figure 8 was as follows: Figure 8A (Line: 429-436), Figure 8B (Line: 512-520), Figure 8C (Line: 527-532), Figure 8D (Line: 532-537).

*In general, it was difficult to relate the discussion section to the corresponding data in the results section. Therefore, for best discussion to the provided data the author is strongly advised to combine the results section and the discussion section.

The Discussion was reorganized, so it follows a similar order as the Results and easier for the readers to relate the discussion to the results.